# PVA-Based Films with Strontium Titanate Nanoparticles Dedicated to Wound Dressing Application

**DOI:** 10.3390/polym16040484

**Published:** 2024-02-09

**Authors:** Beata Kaczmarek-Szczepańska, Lidia Zasada, Marcin Wekwejt, Maria Swiontek Brzezinska, Anna Michno, Anna Ronowska, Magdalena Ciesielska, Ganna Kovtun, M. Teresa Cuberes

**Affiliations:** 1Department of Biomaterials and Cosmetics Chemistry, Faculty of Chemistry, Nicolaus Copernicus University, Gagarin 7, 87-100 Toruń, Poland; 296559@stud.umk.pl (L.Z.); 294291@stud.umk.pl (M.C.); 2Department of Biomaterials Technology, Faculty of Mechanical Engineering and Ship Technology, Gdańsk University of Technology, 80-233 Gdańsk, Poland; marcin.wekwejt@pg.edu.pl; 3Department of Environmental Microbiology and Biotechnology, Faculty of Biological and Veterinary Sciences, Nicolaus Copernicus University in Torun, Lwowska 1, 87-100 Torun, Poland; swiontek@umk.pl; 4Department of Molecular Medicine, Medical University of Gdańsk, 80-210 Gdańsk, Poland; anna.michno@gumed.edu.pl (A.M.); anna.ronowska@gumed.edu.pl (A.R.); 5Institute of Magnetism NAS of Ukraine and MES of Ukraine, Blvd. Acad. Vernadsky 36-b, 03142 Kyiv, Ukraine; ganna.kovtun@uclm.es; 6Department of Applied Mechanics and Project Engineering, Mining and Industrial Engineering School of Almaden, University of Castilla-La Mancha, Plaza Manuel Meca 1, 13400 Almadén, Spain; teresa.cuberes@uclm.es

**Keywords:** hydrogels, polyvinyl alcohol, strontium titanate, nanoparticles

## Abstract

Bioactive materials may be applied in tissue regeneration, and an example of such materials are wound dressings, which are used to accelerate skin healing, especially after trauma. Here, we proposed a novel dressing enriched by a bioactive component. The aim of our study was to prepare and characterize poly(vinyl alcohol) films modified with strontium titanate nanoparticles. The physicochemical properties of films were studied, such as surface free energy and surface roughness, as well as the mechanical properties of materials. Moreover, different biological studies were carried out, like in vitro hemo- and cyto-compatibility, biocidal activity, and anti-biofilm formation. Also, the degradation of the materials’ utilization possibilities and enzymatic activity in compost were checked. The decrease of surface free energy, increase of roughness, and improvement of mechanical strength were found after the addition of nanoparticles. All developed films were cyto-compatible, and did not induce a hemolytic effect on the human erythrocytes. The PVA films containing the highest concentration of STO (20%) reduced the proliferation of *Eschericha coli*, *Pseudomonas aeruginosa*, and *Staphylococcus aureus* significantly. Also, all films were characterized by surface anti-biofilm activity, as they significantly lowered the bacterial biofilm abundance and its dehydrogenase activity. The films were degraded by the compost microorganism. However, PVA with the addition of 20%STO was more difficult to degrade. Based on our results, for wound dressing application, we suggest using bioactive films based on PVA + 20%STO, as they were characterized by high antibacterial properties, favorable physicochemical characteristics, and good biocompatibility with human cells.

## 1. Introduction

The largest organ of the human body is the skin. The skin performs basic functions, such as maintaining the physical barrier, thermoregulation, and participation in the regulation of water loss. Damage to the skin barrier is called a wound. Wound formation can be caused by thermal/physical trauma or a pathological condition [1]. To avoid infection by an external pathogen during skin trauma, tissue regeneration is often accomplished through a skillfully managed healing process that begins soon after the injury occurs [2]. Wound dressings are used to accelerate wound healing and have been a “hot topic” among scientists for several years. While there are countless wound dressings available in the medical market, and scientists are investigating them to enhance their properties to be “ideal” healing materials.

One of the synthetic polymers used in medicine is polyvinyl alcohol (PVA). It is obtained in the saponification process from poly(vinyl acetate) during full or partial hydrolysis. Polyvinyl alcohol is known for its semicrystalline structure. It shows chemical resistance, low carbohydrate adsorption, biodegradability, biocompatibility, and high water solubility [3]. Due to its properties, it is used in various basic areas of everyday life, including medicine, the raw material industry, the food industry (as food packaging), the manufacturing industry, and the paper industry. Due to its film-forming properties [4] and miscibility with other biopolymers/polymers showing hydrophilic properties, PVA is used in medicine to produce wound dressings [5,6], soft contact lenses, eye drops, embolic filters, and artificial cartilage and meniscus [4].

Electrically active materials are attracting increasing interest in biological applications because their unique properties of piezoelectricity, pyroelectricity, and ferroelectricity can be used to promote healing [7,8,9]. Biocompatible polymer nanocomposite films with an electroactive response may be used for physiological deformation sensing, electrostimulation of cells, as smart drug delivery platforms, etc.

Strontium Titanate nanoparticles are perovskites with a high dielectric constant that may exhibit piezoelectricity and flexoelectricity. Recent studies have demonstrated that PVA-SrTiO_3_ nanocomposite films can be prepared with a significant flexoelectric response [10]. The properties of the nanocomposite films have proved to be highly dependent on the nanoparticle concentration, aggregation within the matrix, and the nanoparticle/matrix interface interactions [11]. SrTiO_3_ nanoparticle incorporation in PVA matrices has been considered to improve its dielectric properties [12] and to prepare proton exchange membranes for microbial fuel cells [13]

Many scientists are fascinated by the technology of multilayer titanium dioxide nanotubes. TiO_2_ nanotubes meet the biocompatibility standard and are used to administer drugs [14]. Techniques for modifying nanotubes are widely known, which allow for the preparation of nanotubes compatible with cells. One of the methods of modifying nanotubes is the introduction of strontium (Sr) to titanium, which increases the bioactivity of the surface and positively affects the adhesion, proliferation, and differentiation of pre-osteoblastic cells [15].

For biomedical and biological applications, a thorough investigation of the biocompatibility, nontoxicity, and biodegradability of the materials under consideration is essential. Antibacterial studies on PVA-SrTiO3 nanocomposites indicate a high activity of the films against Gram-positive (*S. aureus*) and Gram-negative (*E. coli*) organisms [16].

The primary objective of this study was to synthesize novel thin-film materials utilizing polyvinyl alcohol impregnated with strontium titanate nanoparticles. The proposed investigations encompass comprehensive assessments, including hemo-compatibility and cyto-compatibility evaluations, analyses of biocidal activity, biofilm characteristics, and the biodegradability of the films in compost. These inquiries provide a nuanced perspective on the potential applications of these materials, specifically as wound dressings, while considering their physicochemical properties and performance in biomedical settings. 

## 2. Materials and Methods

### 2.1. Chemicals

Polyvinyl alcohol (PVA) in granular form (MW 31.000–50.000, 98–99% hydrolyzed) and strontium titanate nanoparticles (NPs) (of about 100 nm in diameter) were purchased from Merck (Darmstadt, Germany). Reagents for in vitro hemo- and cyto-compatibility testing, unless otherwise noted, were purchased from Merck kGaA (Darmstadt, Germany).

### 2.2. Sample Preparation

For the preparation of the polyvinyl alcohol nanocomposite films with strontium titanate nanoparticles (PVA/STO), a 6%wt stock solution of pure PVA was first prepared by consistently stirring PVA granules at 90 °C in distilled water until the PVA was completely dissolved. Next, strontium titanate nanoparticles (STO NPs) were added to the PVA stock solution in different amounts (from 0.5%wt up to 20.0%wt, based on dry PVA weight), and the resulting mixtures were further stirred at 60 °C for about 5 h. Finally, the mixtures were poured into Petri dishes (7 µL in a 90 × 14 mm Petri dish), labeled appropriately, and kept at room temperature for the evaporation of excess water. In about 36 h, PVA/STO films a few microns thick could be easily peeled off from the containers. Due to the addition of STO, the PVA films, transparent in a pure state (in the absence of NPs), acquired a whitish color, more intense in higher STO concentrations.

### 2.3. Surface Free Energy

Surface free energy–IFT(s), polar–IFT(s,P), and dispersive–IFT(s,D) components were calculated by the contact angle measurement of the liquids (glycerin and diiodomethane) by the Owens–Wendt method [17]. The experiment was carried out in a constant temperature value using a goniometer equipped with a drop shape analysis system (DSA 10 Control Unit, Krüss, Germany).

### 2.4. Mechanical Properties

The mechanical properties of films were determined for the elongation test using a testing machine (Shimadzu EZ-Test EZ-SX, Kyoto, Japan). For the measurements, samples of a known thickness were inserted between two handles and stretched (velocity: 5 mm/min). The Young’s modulus was calculated from the slope of the stress–strain curve in the linear region with the use of the Trapezium X Texture program. Each measurement was carried out in ten repetitions.

### 2.5. Roughness

The roughness of the surface of the obtained films was determined from the images obtained by using a microscope with a scanning SPM probe of the NanoScope MultiMode type (Veeco Metrology, Inc., Santa Barbara, CA, USA) operated in a tapping mode. The root-mean-square (Rq) roughness and the arithmetic mean (Ra) were calculated within the Nanoscope v6.11 software (Bruker Optoc GmbH, Ettlingen, Germany).

### 2.6. Swelling

Dry films with a known weight (m_0_) were immersed in distilled water. After 1, 2, 3, 4, and 5 h (m_t_), the subjected samples were removed, gently dried with tissue paper, and weighed. The percentage of the weight change was calculated using equation:$$swelling\;\left[\%\right]=\frac{m_{t}-m_{0}}{m_{0}}\;\times \;100\%$$

### 2.7. Water Content

The water content of the films was measured by drying samples at 105 °C until they reached a constant weight. The results are presented as grams of water per 100 g of a dry sample (n = 5).

### 2.8. In Vitro Hemo- and Cyto-Compatibility Testing

The experiments on in vitro compatibility of the films were conducted on human red blood cells (RBCs) and a human osteoblast cell line (hFOB 1.19, ATCC RRID: CVCL 3708). The number of cells was estimated with a hemocytometer Superior CE (Marienfeld, Lauda-Königshofen, Germany). For hemo-compatibility studies, films with a square shape (5 mm × 5 mm) were used, and for cyto-compatibility studies, their side length was increased to 15 mm. Before testing, all specimens were sterilized by UV-light exposure for 30 min.

RBCs were isolated from buffy coats obtained as by-products of whole blood fractionations from the Regional Blood Centre in Gdańsk (Regional Blood Blank institutional permission M-073/17/JJ/11). Whole blood was collected from healthy volunteers in accordance with the Declaration of Helsinki under an approved Regional Bank review board protocol in standard acid citrate dextrose solutions. The RBCs were fractionated according to the standards of Blood Banks [18]. The RBCs (3 × 10^9^ cells/mL) were placed in 2 mL tubes containing the films and incubated at 37 °C for up to 24 h. The remaining blood samples were centrifuged at 100× *g* at room temperature for 3 min to let the erythrocytes sediment, and supernatants were taken for further research. The hemolysis was assessed spectrophotometrically at a wavelength of 540 nm with an Ultrospect 3000pro spectrophotometer (Amersham-Pharmacia-Biotech, Cambridge, UK). The RBCs treated with 2% Triton were used as a positive control (i.e., 100% hemolysis), and the RBCs incubated without the films were used as a negative control. According to the literature, materials resulting in below 2% of the hemolysis percentage are nonhemolytic [19].

The hFOB cells were grown in a 1:1 mixture of Ham’s F12 Medium and Dulbecco’s Modified Eagle’s Medium (without phenol red), containing 1 mmol/L L-glutamine, 0.3 mg/mL geneticin (G418), and 10% fetal bovine serum at 37 °C in humidified atmosphere containing 5% CO_2_. For the study, extracts from the tested films (n = 4) were used; films were immersed in a 1 mL culture medium for 24 h. The cells were seeded at a density of 12 × 10^3^ cells on a 96-well plate for 24 h, and then the culture medium was changed to those containing film extracts. The cell viability was evaluated after 24 h and 72 h of culture using the MTT assay (3-(4,5-dimethylthiazol-2-yl)-2,5-diphenyltetrazolium bromide). Each well in the culture plate was covered with a fresh medium (100 µL) containing 0.60 mmol/L of MTT, and then incubated for 4 h. The energy metabolism was determined by a spectrophotometric measurement at 570 nm. The results were expressed as a % of change of living cells compared to the results obtained for cells grown on unmodified film of PVA (assumed as 100%). Moreover, the LDH release assay was used to determine the cell death by assessing the level of plasma membrane damage in the tested cell population. The supernatants from the hemo-compatibility test and the culture medium from the cyto-compatibility test were used for this study. The lactate dehydrogenase (LDH, fractional (S)-lactate:NAD+ oxidoreductase) was surveyed by direct measurement of NADH oxidation at 340 nm. LDH data were expressed as a % of the total LDH released from cells.

### 2.9. Biocidal Activity

The biocidal properties of PVA/STO films were studied according to the standard ISO 22196. The study was based on three bacterial oxidants: *Staphylococcus aureus* (ATCC 6538P), *Pseudomonas aeruginosa* (ATCC 4352), and *Escherichia coli* (ATCC 8739). The bacterial strains were grown in Nutrient broth (Biomaxima, Poland) and incubated at 37 °C for 24 h. From each culture, 1 mL of suspension was transferred with a sterile pipette into Eppendorf tubes, which were then centrifuged at 10,000 rpm. After removing the supernatant, the precipitate was suspended in 2 cm^3^ of sterile physiological saline. The obtained bacterial suspension was transferred to a densitometer (Densi-La-Meter^®^ II, Erba Lachema, Czech Republic) for the measurement of its optical density, which was subsequently brought to a value of 0.5. According to McFarland standards, this corresponds to 1.5 × 10⁸ bacterial cells in 1 cm^3^. The suspension was then diluted with sterile physiological saline until the number of bacterial cells was 7.5 × 10⁵ 1 cm^3^. The final suspensions were transferred to PVA/STO films (3 × 3 cm) and covered with sterile glass slides (3 × 3 cm) to evenly distribute the microorganisms on the surface. The samples were incubated at 37 °C for 24 h. The number of viable cells on the surface of the PVA films was determined using the Koch plate method. For this purpose, PVA films were washed with 10 mL sterile saline. The diluted suspension was seeded onto PCA medium (Biomaxima, Poland), and incubation was carried out at 37 °C for 24 h.

The reduction in bacteria cell number was recalculated according to the JIS Z 2801:2012 Standard [20] and expressed as an R-value. The R-value = log (B/C), where B is the viable bacteria cell number in the control sample (PVA) after 24 h of incubation (CFU/mL) and C is the viable bacteria cell number in test samples (PVA/STO) after 24 h of incubation (CFU/mL).

The average of the common logarithm of the number of viable bacteria recovered from the test samples after 24 h were: 5%STO/PVA, 10%STO/PVA, 20%STO/PVA.

To evaluate the effectiveness of the antimicrobial activity, the criteria used by Souli et al. [21] were adopted, according to which a suspension density reduction occurred, ranging from ≤2 to <3 log mean bacteriostatic properties, as well as a reduction of over 3 log bactericidal properties.

### 2.10. Biofilm Analysis

To determine the abundance of biofilm and its dehydrogenase activity, microbiological analyses on the PVA/STO films were conducted. Bacterial biofilm formation on the films was measured using a spectrophotometric method based on the adsorption of crystal violet, as described in previous studies [22,23,24]. The bacterial cultures were pre-incubated in nutrient broth and afterwards transferred onto the films, which were then incubated for 48 h. After incubation, the films were washed and dried. Following that, a 1% solution of crystal violet was applied, and the films were measured the absorbance of the extracted dye using a spectrophotometer. The analysis was performed in triplicate to ensure accuracy and reproducibility.

The dehydrogenase activity of the biofilm formed on the surface of the tested composites was determined using the TTC test. After biofilm formation (described above), the films were rinsed with sterile saline. Then, 3 mL of reaction mixture consisting of TrisHCl buffer pH = 8.4, 2% glucose solution, 0.4% TTC solution, and 0.36% sodium sulphite was applied to the pieces of film (3 × 3 cm) with biofilm. The enzymatic reaction was carried out at 37 °C for 24 h. After incubation, the reaction mixture was decanted from the film, and TF was extracted with n-butanol. The concentration of TF was determined according to the standard curve using a Hitachi U1900 spectrophotometer. The determination was repeated three times.

### 2.11. Biodegradation of PVA/STO Films and Enzymatic Activity in Compost

Biodegradation of PVA/STO films in compost was determined using the OxiTop system, following the modified procedure of Swiontek Brzezinska et al. [25]. To begin, 100 g of compost was placed in a jar and PVA film fragments (1 g) were added. The mixture was then thoroughly mixed and incubated at 26 °C for 21 days. The biodegradability of the PVA film was determined by measuring the oxygen consumption and expressing it in terms of mgO_2_/kg of compost. Additionally, the determination of the enzymatic activity of the compost and its physicochemical properties were included in the study. PVA films with STO can change the activity of enzymes that are involved in polymer degradation. Hence, lipase, aminopeptidase, α-glucosidase, and β-glucosidase activities were included in the study. The activities of these enzymes were determined by the fluorimetric method using a substrate, respectively: MUF-butyrate, MCA-leucine, MUF-α-D-glucoside, and MUF-β-D-glucoside [26,27,28]. A soil sample after 21 days of PVA/STO biodegradation was used for the study. The soil was diluted tenfold, and then a reaction mixture was prepared containing 2 mL of soil extract and 0.2 mL of substrate prepared in phosphate buffer (pH = 7, 50 nmol). The final substrate concentration was 50 µmol. The control was prepared like the studied samples. Prior to substrate addition, the soil extract was inactivated by heating for 10 min at 100 °C. Incubation was carried out in a thermoblock at 40 °C for one hour. After incubation, the concentrations of MUF and MCA released were measured using a Hitachi F-2500 spectrophotometer. For the substrate containing the MUF molecule, the fluorimeter was set at an excitation (EX) wavelength of 318 and an emission (EM) wavelength of 445 nm. For the substrate, MCA-a fluorimeter was set up at an emission wavelength of 345 nm and an excitation wavelength of 425 nm. The amount of µmol of MUF/MCA released per hour was taken as a measure of enzyme activity. The pH of the compost was measured using a glass electrode in 1 M of KCl by the potentiometric method. The total organic carbon (TOC) was assessed using the Thiurin procedure, while the total nitrogen (TN) was determined using the Kjeldahl method [29].

### 2.12. Statistical Analysis

Statistical analysis of the data was performed using commercial software (SigmaPlot 14.0, Systat Software, San Jose, CA, USA). The Shapiro–Wilk test was used to assess the normal distribution of the data. All the results were presented as a mean +/− standard deviation (SD), and were statistically analyzed using one-way analysis of variance (one-way ANOVA). Multiple comparisons versus the control group between means were performed using the Bonferroni *t*-test with the statistical significance set at *p* < 0.05.

## 3. Results and Discussion

### 3.1. Surface Free Energy

It is advisable to carry out modifications of materials to decrease the surface free energy that indicates better integration with surrounding tissues [30]. The addition of STO nanoparticles results in a change in the surface properties of obtained films (Table 1). The surface free energy decreased after the addition of Strontium Titanate (STO) nanoparticles. Also, the polar and dispersion components change after the modification of PVA. A 5% addition of STO increased the polar component and highly decreased the dispersive component. For 10% and 20% addition of STO, the decrease of dispersive component decreased in comparison to 5%STO. Simultaneously, the polar component increased compared to 5%STO content; however, it was lower than in PVA. Similar observations were concluded by Matos de Carvalho et al. [31] for solid lipid nanoparticles entrapped α-tocopherol (up to 50%); however, for the highest concentration (75%), a change in this trend was observed. Also, Popescu reported that PVA films modified with cellulose nanocrystals (up to 15%) showed a decrease in free surface energy [32].

### 3.2. Mechanical Properties

The mechanical properties of the obtained films were determined. The Young Modulus, maximum tensile strength, and the elongation at break are shown in Figure 1. The addition of Strontium Titanate (STO) nanoparticles results in a change in mechanical parameters. PVA-based films without nanoparticles showed higher Young Modulus than films with STO. The addition of STO decreased the stiffness of the films. The 10% and 20% addition of STO improved the maximum tensile strength, but at the same time, a decrease of the elongation at break was observed. The increase in the maximum tensile strength simultaneously with a decrease in elastic modulus of a film containing 10%STO indicates a corresponding significant increase in the ultimate strain. The positive effect of various nanoparticles on the mechanical properties of modified materials was previously confirmed [11]. According to previous UFM results, the PVA/STO interface interactions are very weak; ultrasound is effectively damped at the matrix/nanoparticle interface regions. However, it is strongly related to both the concentration, type, and size of nanoparticles and their distribution in the matrix [33,34,35]. For example, similar results to ours were obtained by Oliveira et al. [36], who reported that the addition of nanosilver led to an improvement in the mechanical strength of PVA-based films.

### 3.3. Roughness

The surface topography of the films has been shown to influence biomaterial–cell interactions. The roughness of the surface changes after the addition of STO nanoparticles to PVA-based film (Table 2). Both parameters rise in tandem with a higher concentration of Strontium Titanate (STO). It suggests that the particles are present on the film surface, and, thereby, that the films are not smooth. The topography of the surface of the films is shown in Figure 2. The addition of nanoparticles generally leads to an increase in the roughness of polymeric films. It was already reported, for example by Lewandowska et al. [37], for the hydroxyapatite nanoparticles incorporated into PVA films.

### 3.4. Swelling

The results of the study of the swelling are shown in Table 3. The PVA-based films dissolved in water. The addition of STO increased their stability. The increase of swelling behavior was observed with the increasing time of immersion in water. However, after 5 h, all films broke down, and the experiment was stopped. The increase in swelling of materials correlated with the rising amount of strontium titanate nanoparticles, which suggests that the nanoparticle/matrix interface interactions are stronger and stabilize the matrix [11]. However, it would be necessary to modify the proposed materials by adding cross-linking agents to improve their stability. The low stability in water conditions could otherwise limit its application as a wound dressing.

### 3.5. Water Content

In the context of wound dressings, the water content of biomaterials is a critical factor that can influence the effectiveness of the dressing in promoting wound healing. The water content in the tested films is shown in Table 4. The results showed that the increasing amount of STO added to the PVA matrix caused the higher water content in material. It suggests that it helps in moisturizing the wound and protecting it from drying out [38].

### 3.6. In Vitro Hemo- and Cyto-Compatibility Testing

The addition of STO into PVA films up to 20% did not negatively affect their cyto-compatibility in a short incubation period (24 h) with prepared extracts, which is confirmed by a comparable MTT result and no increase in the released LDH (Figure 3a). However, longer incubation (72 h) showed that this modification had a slightly cytostatic effect on osteoblast (MTT assay: ~70–80% for PVA with 5% and 10%STO), especially relevant for higher STO content–PVA-20%STO (MTT ~60%) and considering that no significant increase in released LDH into the medium indicates no cytotoxicity (Figure 3b).

All developed films did not have a hemolytic effect on the human erythrocytes (Figure 4), which is confirmed by the low degree of hemolysis (below 0.6%) and the release of LDH from the cells comparable to the negative control.

Strontium titanate nanoparticles may have biomedical applications as carriers for strontium ions, which actively support bone regeneration [39]. So far, various Sr components have been used, among others, in bioactive glasses [40], hydroxyapatite ceramics [41], and as a coating for titanium surfaces [42]. Moreover, Sr-containing materials hemo- and cyto-compatible were previously confirmed [43,44]. Furthermore, a number of studies, both in vitro and in vivo, have also shown that PVA is a highly biocompatible material [45]. Hence, the non-toxicity of our composites based on PVA with STO could be assumed, and improved cellular response could also be expected. However, results of hFOB compatibility consistent with the literature were obtained only for a shorter culture period (24 h), where the higher cell viability (with a positive trend with increasing STO concentration) compared to the unmodified control was found (up to 10%STO). Nevertheless, with a longer incubation period (72 h), the opposite effect was observed. Such a positive effect of PVA on the viability and proliferation of bone-like cells has been previously reported [46]. However, for PVA-STO, especially 20%, a cytostatic effect was noted (no increase in the LDH level was observed), which may be related to the material degradation during the cell culture. It was noticed in the literature that most of the research on the modification of biomaterials with strontium components did not use it in the nanometric form (such as in our work, ~100 nm diameter), which may also partially explain the differences in the obtained results. Further, it is also worth emphasizing that all films remained hemo-compatible, which is fully consistent with the reports of other researchers [43]. Therefore, based on the ISO 10993-5 standard [47] and the analysis of statistical significance, PVA film containing up to 10%STO may be considered biocompatible.

### 3.7. Biocidal Properties of PVA/STO Films

Studies on the effect of adding STO to PVA films showed that films containing the highest amount of STO (20%) reduced the abundance of test strains most strongly. When analyzing the values for reduction in bacterial abundance on the films tested, no significant differences were observed between the strains (Figure 5).

Among various types of existing materials, polyvinyl alcohol (PVA) is widely applied for fabrication of antibacterial materials as it is a non-toxic, biocompatible polymer displaying chemical stability and excellent film-forming properties [48,49,50]. Antimicrobial properties of nanocomposites (PVA-PAA-SrTiO_3_) using *Staphylococcus aureus* and *Escherichia coli* bacteria were studied by Ghalib et al. [16]. They fabricated a film from pure polyvinyl alcohol (PVA), a mixture of polyacrylic acid (PAA), and a PVA/PAA mixture doped with strontium titanate (SrTiO_3_ NPs). They used a diffusion method to determine the biocidal properties. The authors showed that the zone of inhibition becomes larger with increasing concentrations of nanoparticles. The potential mechanism of action is that the nanocomposites (PVA-PAASrTiO_3_) have positive charges, while the bacteria have negative charges, resulting in an electromagnetic attraction between the nanocomposite nanoparticles and the microorganisms. The microorganisms are oxidized and die as soon as the attraction is achieved [51]. In contrast, Zhang et al. [52] investigated strontium titanate metal oxide (SrTi1-xFexO_3_- or STFx for short) as an effective antibacterial agent. They tested the biocidal properties using *Escherichia coli* (*E. coli*) in the presence of dispersed STF0.8 nanoparticles in water. The authors showed an excellent bactericidal effect by killing all *E. coli* (100%) within 15 min. The biocidal properties in the presence of STO were also tested. However, they did not observe a biocidal effect on *E. coli*. In our study, we observed a significant reduction in the abundance of *S. aureus*, *E. coli*, and *P. aeruginosa* only on PVA films with the addition of 20%STO. However, according to Souli et al. [53], a reduction of two orders of magnitude in the number of cells capable of growth (R > 2) is interpreted as a biocidal effect of the tested films. Our study shows that the reduction in the number of bacteria tested was R = 1.9.

### 3.8. Quantitative and Enzymatic Biofilm Analysis

The biofilm forming on the PVA/STO surface was studied in terms of biofilm abundance and respiratory activity (Figure 6). Bacteria formed biofilm on the surface of the films tested in varying abundance. *Staphylococcus aureus* and *Pseudomonas aeruginosa* formed a biofilm of similar abundance. However, *Staphylococcus aureus*’ biofilm abundance was found to be significantly higher on PVA film than on PVA/STO. In contrast, *Pseudomonas aeruginosa* formed a strong biofilm on the PVA surface and PVA5–10%STO. Only on the surface of PVA20%STO was a lower biofilm abundance observed. *E. coli* formed a biofilm on the surface of PVA films with lower abundance. The biofilm abundance was significantly lower on the surface of PVA films with 5, 10, and 20%STO than on PVA films. When analyzing respiratory activity values, a decrease in activity was observed for all strains on the surface of PVA films with 5, 10, and 20%STO.

The biocidal activity of films can also be determined by biofilm formation on the polymer surface and its respiratory activity. Our study revealed a significant reduction in both biofilm abundance and its respiratory activity exclusively in PVA films with the addition of 20%STO compared to PVA films and PVA5–10%STO. The abundance of biofilm, as well as its enzymatic activity, depends primarily on the type of film and the physiological state of the bacteria. *S. aureus* and *P. aeruginosa* show intense biofilm formation across various polymer surfaces. Most infections, especially hospital-acquired infections, are significantly associated with the strong, biofilm-forming *Pseudomonas aeruginosa* and *Staphylococcus aureus* [54]. Wu et al. [55] prepared mixed films based on poly(vinyl alcohol) (PVA) and chitosan (CH). According to the authors, CH60:PVA40 film showed significant activity against adhesion and inhibited the biofilm formation of *P. aeruginosa* PAO1, indicating that CH60:PVA40 film can be used as a food packaging material with antimicrobial activity and inhibition of biofilm formation.

### 3.9. Biodegradation of PVA/STO Films in the Compost

Biodegradation of PVA with STO was determined by the oxygen consumption of the microorganisms inhabiting the compost (Figure 7). The study showed that PVA film and PVA with the STO addition were degraded by the compost microorganisms proportionally with the duration of incubation. After 21 days of incubation, the microorganisms required 225 mgO_2_/kg to degrade the PVA film. The PVA film with the STO addition was not significantly less degraded in the compost. In contrast, the microorganisms consumed 150 mgO_2_/kg to decompose PVA with the addition of 20%STO.

Determining the biodegradability of polymers in the environment is crucial since polymers have varying degrees of degradation. Undoubtedly, microorganisms producing enzymes have a significant impact on the degradation of polymers. The biodegradation of polymers takes place in the natural environment, e.g., in soil, or in anthropogenic environments, e.g., in compost. In our study, mature compost was selected, and biodegradation determinations were determined by the use of oxygen in a closed vessel. The study showed that PVA films with the addition of 20%STO were noticeably less degraded by the compost’s microorganisms. It is most likely that the addition of 20%STO to the PVA film reduces the viability of compost microorganisms. Biocidal studies of PVA/STO have shown that 20%STO significantly reduces the abundance of *E. coli*, *S. aureus* and *P. aeruginosa*. Despite the results, a complete inhibition of the growth of these bacteria was not observed. In compost, the presence of microbial biodiversity may favor the decomposition of the polymers tested. However, this process requires a significant amount of time. OxiTop allows oxygen consumption results to be recorded in the presence of polymers and has its limitations. For longer incubations, oxygen consumption is not observed due to oxygen consumption depletion by microorganisms. Chiellini et al. [56] reported that poly(vinyl alcohol) (PVA) is considered one of the few vinyl polymers susceptible to biodegradation in the presence of suitably acclimatized microorganisms. Consequently, more and more attention is being paid to the preparation of environmentally friendly PVA-based materials for a wide range of applications. However, it should be taken into account that the introduction of antimicrobial substances into PVA may inhibit biodegradation by microorganisms. Zhang et al. [52] reported that strontium titanate had a weak biocidal effect on *E. coli*. Strong biocidal activity was shown by strontium titanate ferrite. In turn, Gao et al. [57] reported that zirconium-doped strontium titanate (AZSTO) nanofibrous membranes showed potent bactericidal activity against *S. aureus* and *E. coli*. Furthermore, the biodegradability of PVA depends on hydrolysis and its molecular weight [58]. Bonilla et al. [59] studied the biodegradability of natural gelatin (GEL) and sodium caseinate (SCas) and synthetic poly(vinyl alcohol) (PVA) and poly(lactic acid) (PLA) biopolymers films using an aquatic system from active sludge. Biodegradation was determined by measuring the dissolved oxygen over 28 days with closed respirometers. According to the authors, GEL and SCas films were completely biodegraded on day one. The PVA and PLA films required more time to biodegrade. Their results showed that PVA is not a biodegradable polymer. However, the degradation of PVA occurs faster than that of PLA.

### 3.10. Effect of PVA Films with STO on the Change in Enzyme Activity in Compost

The introduction of PVA film to the compost did not change the activity of the extracellular enzymes responsible for the degradation of organic matter (Table 5). Compared to the control (compost without PVA film), the activity of the enzymes tested were similar. In contrast, the addition of 10 and 20%STO to PVA film significantly reduced the activity of the enzymes: lipase, aminopeptidase and β-glucosidase. The activity of α-glucosidase in the compost was similar when all films were introduced into the compost.

The introduction of polymers into the environment can affect microorganisms and degradation processes differently. The addition of antimicrobial substances to polymer films can alter the abundance and extracellular enzyme release by them. A reduction in the activity of enzymes that are responsible for the degradation of organic matter is negative for the environment. Our study showed that only PVA film with 20%STO significantly decreases extracellular enzyme activity. Enzyme activity tests can be a good indicator of overall microbial activity. The enzymes we tested are not directly involved in the biodegradation of PVA; however, their determination allows us to determine whether the presence of PVA in the environment may interfere with the degradation of other macromolecular compounds such as proteins or fats. Sakai et al. [60] showed that microbial degradation of PVA in aqueous media by oxidase and dehydrogenase led to the formation of b-hydroxy ketone and 1,3-diketone, which were further degraded by a specific b-diketone hydrolase to form carboxyl and methyl ketone end groups. Kawai and Hu [61] investigated different aspects of microbial degradation of PVA. PVA degradation depends on the 1,2-diol content, ethylene content, tacticity, degree of polymerization, and degree of saponification of the main chain. Sameer [62] reported that PVA showed a slower degradation rate with amylase than PVA/HPP composites. HPP (hemp protein particles) in PVA showed a higher degree of enzymatic degradation (95.5%), while in the case of pure PVA it was 67.5%.

## 4. Conclusions

Effective wound care and treatment remains a persistent challenge. An effective strategy to solve this issue might be to use a bioactive modification for synthetic films, which are characterized by favorable physicochemical properties. Here, we enriched PVA-based films with strontium titanate nanoparticles, which are characterized by high bioactive effects, both in terms of tissue regeneration and antibacterial properties. The addition of nanoparticles improved the surface roughness of films and their mechanical parameters, but contributed to lowering free surface energy. The modification did not negatively affect the biocompatibility of the films tested on the selected human cells and human erythrocytes. The PVA films containing 20%STO showed good antibacterial properties against *Escherichia coli*, *Pseudomonas aeruginosa*, and *Staphylococcus aureus*, as well as being characterized by anti-biofilm activity. The biodegradation of PVA with higher STO concentration was slower. Due to the promising results obtained here, future research will concern a more detailed characterization of the bioactive effect of the film modifications on skin regeneration and more efficient degradation. We believe that the proposed films have future potential for use as bioactive dressings, coatings for implants, e.g., in dentistry or orthopedics, or as packaging materials.

## Figures and Tables

**Figure 1 polymers-16-00484-f001:**
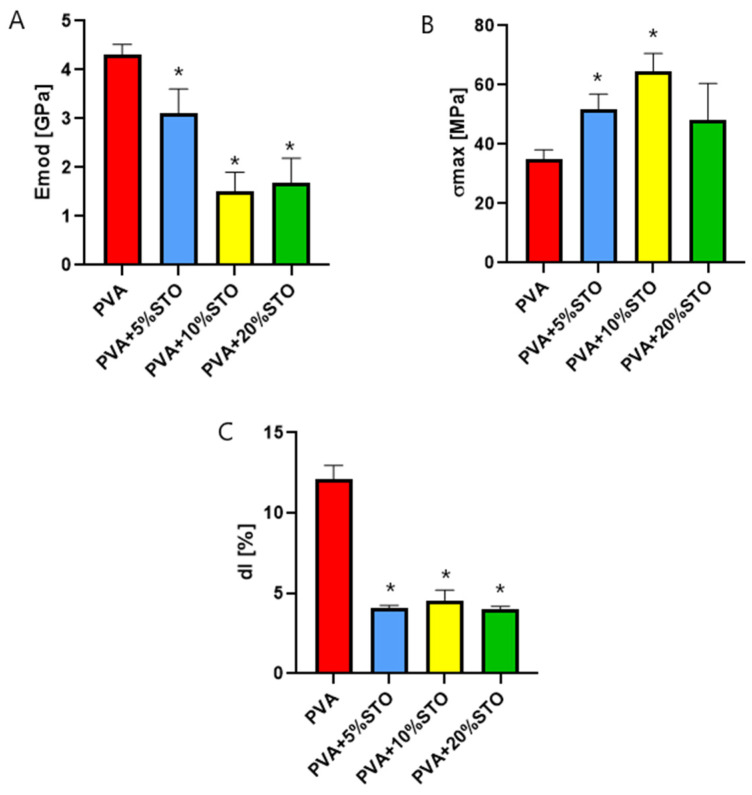
(**A**) the Young Modulus (E_mod_), (**B**) maximum tensile strength (σ_max_) and (**C**) elongation at break (dl) determined for films based on PVA, PVA + 5%STO, PVA + 10%STO, and PVA + 20%STO. * significantly different from PVA (*p* < 0.05).

**Figure 2 polymers-16-00484-f002:**
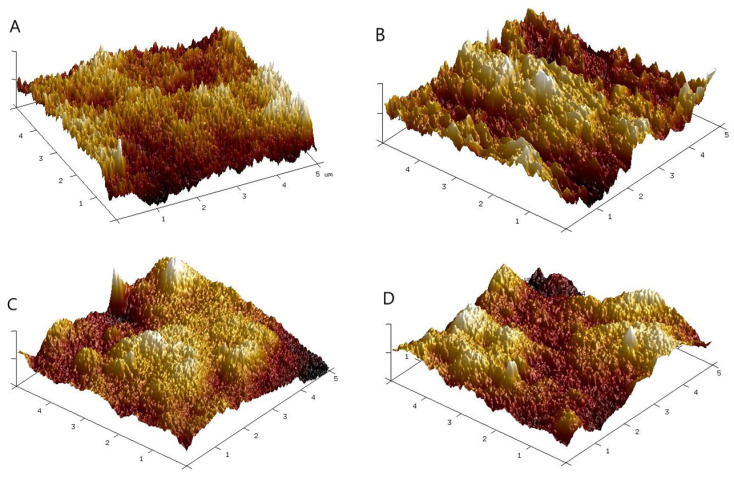
3D images of films surface (**A**) PVA; (**B**) PVA + 5%STO; (**C**) PVA + 10%STO; (**D**) PVA + 20%STO.

**Figure 3 polymers-16-00484-f003:**
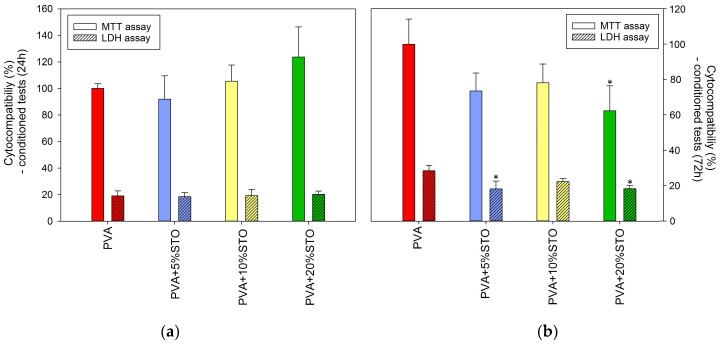
The effect of developed films on cyto-compatibility of hFOB 1.19 cells (cell viability and lactate dehydrogenase release): (**a**) after 24 h and (**b**) after 72 h exposure to films extracts (n = 4; data are expressed as the mean ± SD, * statistical significance compared to the control–PVA (*p* < 0.05)).

**Figure 4 polymers-16-00484-f004:**
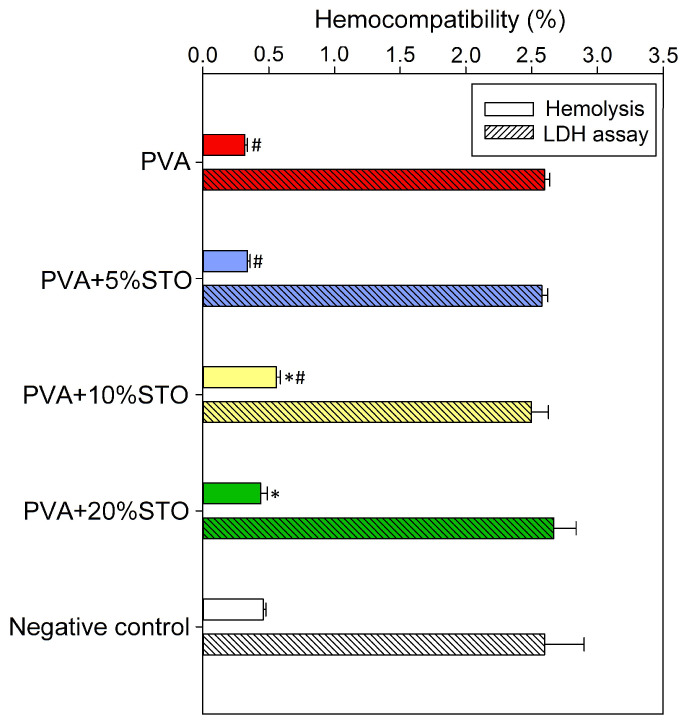
The effect of developed films on hemo-compatibility of human erythrocytes (hemolysis rate and lactate dehydrogenase release) after 24 h exposure to films (n = 4, data are expressed as the mean ± SD, * significantly different from the negative control and ^#^ significantly different from the PVA (*p* < 0.05)).

**Figure 5 polymers-16-00484-f005:**
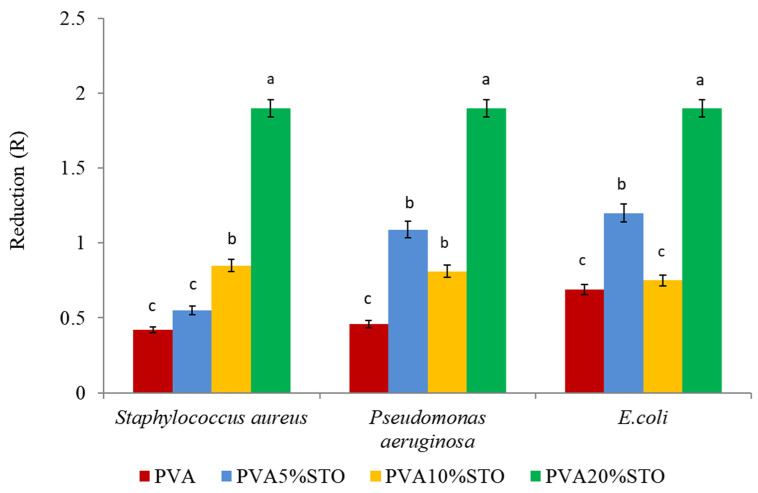
Biocidal properties of PVA/STO films. Different letters over the bars indicate a significant difference between means (*p* < 0.05).

**Figure 6 polymers-16-00484-f006:**
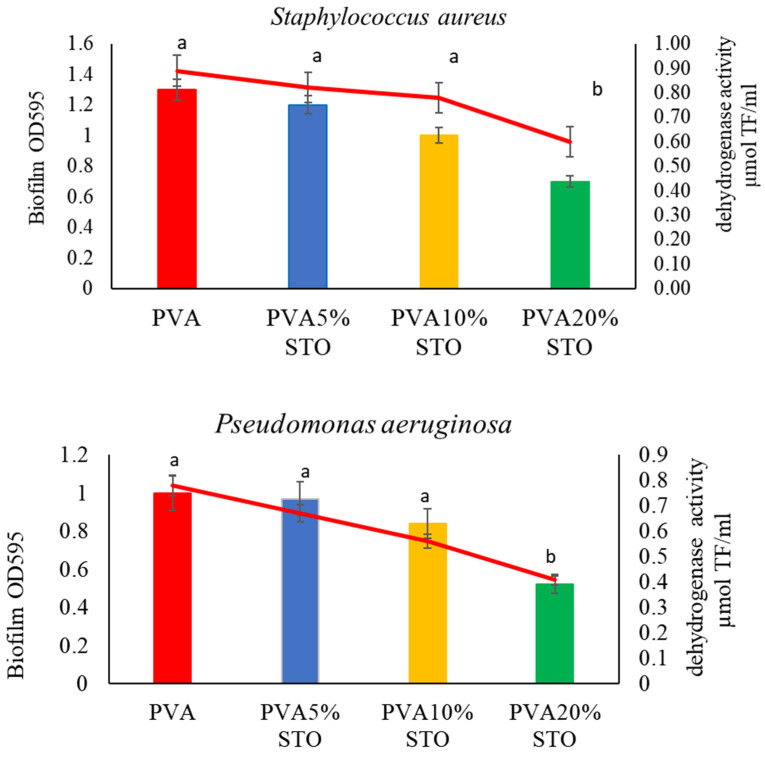
Biofilm abundance and respiratory activity on the surface of PVA/STO films. Columns were designated as biofilm and lines as dehydrogenase. Different letters over the bars indicate a significant difference between means (*p* < 0.05), ±SD (n = 3).

**Figure 7 polymers-16-00484-f007:**
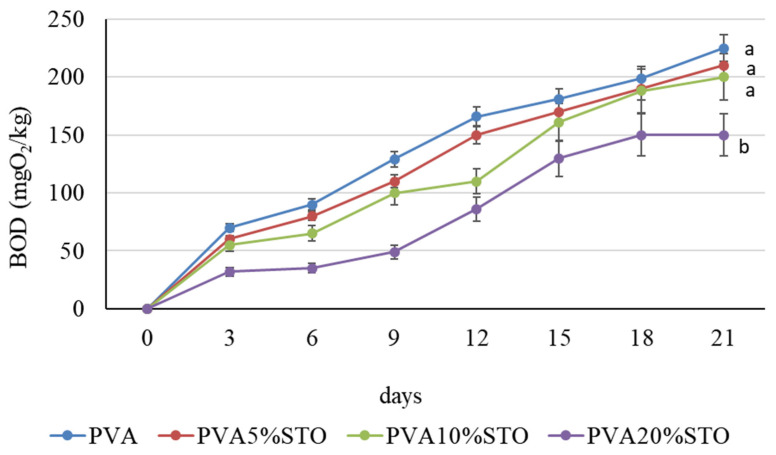
Biodegradation of PVA/STO as expressed in mgO_2_/kg compost. Different letters over the bars indicate a significant difference between means (*p* < 0.05), ±SD (n = 3).

**Table 1 polymers-16-00484-t001:** The surface free energy (IFT (s)), its polar (IFT (s, P) and dispersive (IFT (s, D)) component of PVA films with and without STO (n = 5; * significantly different from PVA (*p* < 0.05)).

Specimen	IFT(s) [mJ/m^2^]	IFT(s,D) [mJ/m^2^]	IFT(s,P) [mJ/m^2^]
PVA	43.69 ± 0.84	33.57 ± 0.48	10.12 ± 0.36
PVA + 5%STO	41.77 ± 0.93 *	39.61 ± 0.70 *	2.16 ± 0.23 *
PVA + 10%STO	39.74 ± 0.91 *	32.66 ± 0.65 *	7.07 ± 0.26 *
PVA + 20%STO	34.76 ± 1.17 *	31.50 ± 0.73	9.27 ± 0.45

**Table 2 polymers-16-00484-t002:** Roughness parameters (Ra and Rq) of films (n = 5; * significantly different from PVA (*p* < 0.05)).

Specimen	Ra [nm]	Rq [nm]
PVA	1.75 ± 0.09	2.22 ± 0.04
PVA + 5%STO	1.83 ± 0.06	2.27 ± 0.03
PVA + 10%STO	2.22 ± 0.02 *	2.80 ± 0.07 *
PVA + 20%STO	2.47 ± 0.03 *	3.25 ± 0.02 *

Ra—mean arithmetic deviation of the profile from the mean line; Rq—mean square deviation of surface roughness.

**Table 3 polymers-16-00484-t003:** The swelling of films of PVA with 5, 10, and 20% of STO addition (n = 5; * significantly different from PVA + 5%STO (*p* < 0.05); ^#^ significantly different from PVA + 10%STO (*p* < 0.05)).

Specimen	Swelling [%]				
	1 h	2 h	3 h	4 h	5 h
PVA	dissolved				
PVA + 5%STO	627 ± 11	691 ± 19	821 ± 17	883 ± 12	915 ± 14
PVA + 10%STO	719 ± 22 *	782 ± 13 *	844 ± 11	909 ± 19	922 ± 17
PVA + 20%STO	740 ± 17 *	798 ± 11 *	901 ± 14 *^#^	933 ± 15 *	942 ± 20 *

**Table 4 polymers-16-00484-t004:** The water content in films based PVA with and without STO (grams of water per 100 g of a dry sample, n = 5; * significantly different from PVA—*p* < 0.05).

Specimen	Water Content [g/100 g]
PVA	3.72 ± 1.05
PVA + 5%STO	6.09 ± 0.91 *
PVA + 10%STO	7.25 ± 0.77 *
PVA + 20%STO	9.17 ± 1.17 *

**Table 5 polymers-16-00484-t005:** Effect of PVA films with STO on the change in enzyme activity in compost.

Films	Enzyme Activity (µM/h)			
	Lipase	α-glucosidase	β-glucosidase	Aminopeptidase
Compost-control	37.2 ± 0.11 ^a^	11.2 ± 0.21 ^a^	25.6 ± 0.33 ^a^	40.0 ± 0.32 ^a^
PVA	36.5 ± 0.12 ^a^	10.5 ± 0.41 ^a^	24.2 ± 0.62 ^a^	38.7 ± 0.22 ^a^
PVA + 5%STO	23.3 ± 0.24 ^b^	9.6 ± 0.16 ^a^	24.4 ± 0.15 ^a^	37.4 ± 0.13 ^a^
PVA + 10%STO	23.4 ± 0.31 ^b^	10.0 ± 0.18 ^a^	12.9 ± 0.48 ^b^	26.5 ± 0.23 ^b^
PVA + 20%STO	10.1 ± 0.10 ^c^	9.3 ± 0.12 ^a^	11.6 ± 0.11 ^b^	26.1 ± 0.26 ^b^

Values are expressed as mean ± SD (n = 3). Different letters in the column indicate statistically significant differences at *p* < 0.05.

## Data Availability

Data are contained within the article.

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
