# Peer review of "PVA-Based Films with Strontium Titanate Nanoparticles Dedicated to Wound Dressing Application"

_polymers, 2024, doi:10.3390/polym16040484_

Round 1

Reviewer 1 Report

Comments and Suggestions for Authors

The research article titled, “PVA-based Films with Strontium Titanate Nanoparticles Dedicated to Wound Dressing Application”, describes the fabrication of Strontium Titanate nanoparticles dispersed polymer films, which could be used be potentially used in wound dressing application. Although PVA- Strontium titanate nanoparticles are not new, as indicated by numerous references by authors itself, this work analyses the various effects such as mechanical property, hemocompatibility, cytocompatibility, antibacterial activity, biodegradation altogether. The work would be interesting to various readers, especially researchers working on antibacterial films and membranes.  However, the manuscript needs some major revisions, before publication.

1.      Page 3, Line 99, indicate the full form of the abbreviations where it first appears.

2.      Section 2.2, How much of precursor solution was poured into which size of the Petri dish. And what was the thickness of the resulting films.

3.      Figure 3, The text in the figures should be made more legible. (Applicable to most figures)

4.      Figure 3, the figure legend indicates a & b panel. Kindly label the panels in the figure as well.

5.      Line 338-362, the authors discuss the probable effect of Strontium ions that are being released. The authors could carry out ion release profile of Sr and Ti from the films (at least for 7 days), which would be confirmatory data for this discussion.

6.      The authors should follow a similar color code for the samples in all the figures, so that it will be easy for the readers to follow.

7.      Since the films are intended to be used as wound dressing, the authors could measure the film swelling (in water) too, as it would give an idea of how much exudates it can absorb.

Comments on the Quality of English Language

Minor grammatical errors need to be checked. It would be better to use a decimal point( eg 0.5, 20.0) rather than a comma (e.g. 0,5 , 20,0).

Author Response

Dear All,

on behalf of myself and co-authors, I am enclosing the manuscript polymers-2801594  entitled “PVA-based films with strontium titanate nanoparticles dedicated to wound dressing application” that we believe should be of strong interest to the general readership of the Polymers journal.

We would like to note that in addition to addressing all reviewer’s valuable remarks, the authors placed additional editorial corrections including references to improve the quality of the manuscript. Below are our point-by-point responses to reviewer’s comments:

Reviewer #1:

  1. Page 3, Line 99, indicate the full form of the abbreviations where it first appears.

Thank you for this comment. It is now corrected.

  1. Section 2.2, How much of precursor solution was poured into which size of the Petri dish.

Thank you very much. It is now corrected:

“90 x 14 mm Petri dishes were typically used, in which 7 µl of the PVA stock solution mixed with the STO nanoparticles was poored and kept at room temperature for the evaporation of excess water.  Films with thickness of a few microns were obtained.

  1. Figure 3, The text in the figures should be made more legible. (Applicable to most figures)

Thank you very much. Figure 3 is now corrected.

  1. Figure 3, the figure legend indicates a & b panel. Kindly label the panels in the figure as well.

Thank you very much. Figure 3 is now corrected.

  1. Line 338-362, the authors discuss the probable effect of Strontium ions that are being released. The authors could carry out ion release profile of Sr and Ti from the films (at least for 7 days), which would be confirmatory data for this discussion.

Thank you very much for the valuable comment. We could not study the ions release as films were not stable for 7 days. After 24 hours they broke down. The study of films’swelling/degradation is discussed in section 3.4. Therefore, we changed also the discussion part in section 3.6.

  1. The authors should follow a similar color code for the samples in all the figures, so that it will be easy for the readers to follow.

Thank you for this comment. It is now corrected.

  1. Since the films are intended to be used as wound dressing, the authors could measure the film swelling (in water) too, as it would give an idea of how much exudates it can absorb.

Thank you very much for this comment. We added the study of swelling behaviour of films (section 2.6. and 3.4).

Reviewer 2 Report

Comments and Suggestions for Authors

The article presented can be of interest to the readers of the journal.

But there are some questions and notes to arise while reading the manuscript.

1. What is the water content of the films prepared by the authors?

Does the presence of Strontium Titanate nanoparticles (NPs) in the polymer and their concentration affect this characteristic?

2. It is not clear for what reason, characterizing the mechanical properties of the studied materials, the authors do not give strength values, but absolute values of the forces expended to break the films. After all, the authors know the cross-sectional areas of the studied samples (otherwise it would not have been possible to calculate the values of the Young’s modulus), so the strength values, the standard mechanical characteristic of the material, could be calculated without difficulty.

3. It is reasonable to comment and discuss the mechanical characteristics of the tested materials presented by the authors. It is obvious that the decrease in the elastic modulus observed as a result of the introduction of nanoparticles into the polymer matrix indicates a weakening of the system of interchain interactions in the material and, consequently, a low degree of compatibilization of the components of the obtained composites, the absence of the interactions between the nanoparticles and the polymer chains (maybe – the increased porosity at the interphase borders). Is this effect a fundamental characteristic of the PVA-nanoSTO system, or does it indicate certain technological problems that could not be solved during the fabrication of the material?

4. Elasticity is important for materials used as soft wound dressings. At the same time, the article does not contain any information about this characteristic of the studied materials. It is reasonable to give data on the ultimate deformations of films, especially since this characteristic obviously demonstrates a specific dependence on the concentration of nanoparticles in the material. Indeed, a sharp increase in the maximum elongation force simultaneously with a drop in the elastic modulus of a film containing 10% STO means that there is an equally sharp increase in the ultimate strain. This nontrivial effect deserves a mention in the paper.

5. Rows 298-300:
“Both parameters rise in tandem with higher concentration of Strontium Titanate (STO). It suggests that the particles are present on the film surface and thereby, films are not smooth.”
It is unclear how an increase in the depth of relief of the film surface by fractions of a nanometer (maximum increase - by ~1 nm when introducing 20% STO into the polymer, see the Table 2) can indicate the presence of STO particles of 100 nm on this surface (of about 100 nm in diameter” – row 94)!

6. In Figure 4, the designation "hemocompatibility, %" refers to the horizontal axis, not the vertical axis as in the article.

7. The experimental results obtained by the authors are rather difficult to compare with each other because of the difference in the state in which the films were in during these experiments. Indeed, the physical characteristics of the films were determined on dried samples, as follows from the description of the methodology of their preparation (however, as noted above, the real moisture concentration in them after drying at room temperature is unknown). At the same time, all experiments of the "biological" block of studies were carried out on forcibly moistened samples (and the water content in them after such moistening, again, is unknown). At the same time, it is known that the presence of water in PVA films significantly affects the properties of the material. Finally, it is obvious that the use of the material for its intended purpose - as soft wound dressings - presupposes its moistening. Apparently, the authors should at least be more rigorous in characterizing the water content in the study objects in different experiments described in the article.

Author Response

Dear All,

on behalf of myself and co-authors, I am enclosing the manuscript polymers-2801594  entitled “PVA-based films with strontium titanate nanoparticles dedicated to wound dressing application” that we believe should be of strong interest to the general readership of the Polymers journal.

We would like to note that in addition to addressing all reviewer’s valuable remarks, the authors placed additional editorial corrections including references to improve the quality of the manuscript. Below are our point-by-point responses to reviewer’s comments:

Reviewer #2:

  1. What is the water content of the films prepared by the authors? Does the presence of Strontium Titanate nanoparticles (NPs) in the polymer and their concentration affect this characteristic?

 Thank you for the comment. The water content of the films was now considered. Indeed, the concentration of Strontium Titanate nanoparticles (NPs) in the film affects it. It is presented and discussed in section 3.5.

  1. It is not clear for what reason, characterizing the mechanical properties of the studied materials, the authors do not give strength values, but absolute values of the forces expended to break the films. After all, the authors know the cross-sectional areas of the studied samples (otherwise it would not have been possible to calculate the values of the Young’s modulus), so the strength values, the standard mechanical characteristic of the material, could be calculated without difficulty.

 Thank you very much for the comment. We added the tensile strength values (section 3.2).

  1. It is reasonable to comment and discuss the mechanical characteristics of the tested materials presented by the authors. It is obvious that the decrease in the elastic modulus observed as a result of the introduction of nanoparticles into the polymer matrix indicates a weakening of the system of interchain interactions in the material and, consequently, a low degree of compatibilization of the components of the obtained composites, the absence of the interactions between the nanoparticles and the polymer chains (maybe – the increased porosity at the interphase borders). Is this effect a fundamental characteristic of the PVA-nanoSTO system, or does it indicate certain technological problems that could not be solved during the fabrication of the material?

     Thank you very much for this comment. According to previous UFM results (ref. 11 of the submitted version) the PVA/STO interface interactions are indeed very weak; ultrasound is effectively damped at matrix/nanoparticle interface regions. It is now added in the discusson part.

  1. Elasticity is important for materials used as soft wound dressings. At the same time, the article does not contain any information about this characteristic of the studied materials. It is reasonable to give data on the ultimate deformations of films, especially since this characteristic obviously demonstrates a specific dependence on the concentration of nanoparticles in the material. Indeed, a sharp increase in the maximum elongation force simultaneously with a drop in the elastic modulus of a film containing 10% STO means that there is an equally sharp increase in the ultimate strain. This nontrivial effect deserves a mention in the paper.

 Thank you very much for the valuable comment. We added the elongation at break values and indeed, we observed change of the parameter values after addition of nanoparticles related to their concentration.

  1. Rows 298-300:

“Both parameters rise in tandem with higher concentration of Strontium Titanate (STO). It suggests that the particles are present on the film surface and thereby, films are not smooth.”

It is unclear how an increase in the depth of relief of the film surface by fractions of a nanometer (maximum increase - by ~1 nm when introducing 20% STO into the polymer, see the Table 2) can indicate the presence of STO particles of 100 nm on this surface (“of about 100 nm in diameter” – row 94)!

Thank you for this comment.  The thickness of our films was in the range 0.087-0.092 mm. Thereby, the addition of nanoparticles sized about 100 nm in diameter did not affect high change of roughness due to their immersion in polymeric matrix.

  1. In Figure 4, the designation "hemocompatibility, %" refers to the horizontal axis, not the vertical axis as in the article.

Thank you for comment. It is now corrected.

  1. The experimental results obtained by the authors are rather difficult to compare with each other because of the difference in the state in which the films were in during these experiments. Indeed, the physical characteristics of the films were determined on dried samples, as follows from the description of the methodology of their preparation (however, as noted above, the real moisture concentration in them after drying at room temperature is unknown). At the same time, all experiments of the "biological" block of studies were carried out on forcibly moistened samples (and the water content in them after such moistening, again, is unknown). At the same time, it is known that the presence of water in PVA films significantly affects the properties of the material. Finally, it is obvious that the use of the material for its intended purpose - as soft wound dressings - presupposes its moistening. Apparently, the authors should at least be more rigorous in characterizing the water content in the study objects in different experiments described in the article.

Thank you very much for this valuable comment. The water content analysis was carried out and we confirmed that proposed films may help moisture the wound from drying out.

Round 2

Reviewer 1 Report

Comments and Suggestions for Authors

The authors have carried out all the possible revisions as suggested. The manuscript can be accepted in its current form. However, the authors are suggested to mention a few lines about the major drawback of the developed wound dressing film and how it could be overcome in the discussions, i.e., breakage within 5 hours of contact with fluids, which has to be improved in the future, because a wound dressing cannot be changed several times in a day.

Comments on the Quality of English Language

Not Applicable.

Author Response

Reviewer #1:

The authors have carried out all the possible revisions as suggested. The manuscript can be accepted in its current form. However, the authors are suggested to mention a few lines about the major drawback of the developed wound dressing film and how it could be overcome in the discussions, i.e., breakage within 5 hours of contact with fluids, which has to be improved in the future, because a wound dressing cannot be changed several times in a day.

Thank you very much for the valuable comment. We added comment:

“However, it would be necessary to modify the proposed materials by adding cross-linking agents to improve their stability. The low stability in water conditions could otherwise limit its application as a wound dressing.”